# Comparison of *Origanum* Essential Oil Chemical Compounds and Their Antibacterial Activity against *Cronobacter sakazakii*

**DOI:** 10.3390/molecules27196702

**Published:** 2022-10-08

**Authors:** Xiaoqi Guo, Yuanpeng Hao, Wenying Zhang, Fei Xia, Hongtong Bai, Hui Li, Lei Shi

**Affiliations:** 1Key Laboratory of Plant Resources and Beijing Botanical Garden, Institute of Botany, Chinese Academy of Sciences, Beijing 100093, China; 2University of Chinese Academy of Sciences, Beijing 100049, China

**Keywords:** essential oil, chemical composition, thymol, antibacterial activity

## Abstract

*Origanum vulgare* L. (oregano) is an aromatic plant with wide applications in the food and pharmaceutical industries. *Cronobacter sakazakii*, which has a high detection rate in powdered infant formula, adversely impacts susceptible individuals. Oregano essential oil (OEO) is a natural antibacterial agent that can be used to fight bacterial contamination. Here, OEO chemical compounds from eight oregano varieties were analyzed by gas chromatography–mass spectrometry and their antibacterial properties were assessed. The eight OEOs were clustered into two groups and were more diverse in group 2 than in group 1. Six compounds, including p-cymene, 3-thujene, γ-terpinene, thymol, carvacrol, and caryophyllene, were shared by eight OEOs. Among the eight oregano varieties, OEOs from *O. vulgare* sc2 had the strongest antibacterial activity against *C. sakazaki*, with the inhibition zone of 18.22mm. OEOs from *O. vulgare* jx, *O.* ‘Nvying’, *O. vulgare* ‘Ehuang’, and *O. vulgare* ssp. *virens* were also potent. Moreover, the antibacterial activity of OEOs was positively correlated with the relative content of thymol. As the main OEO antibacterial compound, thymol affected the normal growth and metabolism of *C. sakazakii* cells by destroying the bacterial membrane and decreasing the intracellular ATP concentration. Thus, in light of the antibacterial activity detected in the OEOs from the eight oregano varieties, this study provides a theoretical foundation for oregano cultivar management and development.

## 1. Introduction

Natural plant extracts, such as essential oils and organic acids, have a specific antibacterial effect [1]. Essential oil exerts a bacteriostatic effect on multiple targets in bacteria by disrupting the bacterial membrane and reducing the intracellular ATP concentration and intracellular pH value [2,3]; thus, developing antibiotic resistance is not easy. In addition, essential oil has been extensively studied for applications in food, hygiene, cosmetics, and medicine for its antibacterial properties [4,5,6].

*Origanum vulgare* L. (oregano), a perennial aromatic plant that belongs to the family Lamiaceae, is commonly used as a flavoring herb, feed additive, and garden ornamental, but the plant is also medicinal and has antibacterial, antioxidant, and anti-inflammatory properties [7]. Besides tremendous application potential in food preservation because of its rich active compounds, such as carvacrol and thymol [8], oregano has a specific bacteriostatic effect on *Escherichia coli* [9], *Staphylococcus aureus* [10], *Listeria monocytogenes* [11], *Salmonella enteritidis* [12], and *Cronobacter sakazakii* [13]. 

The oregano essential oil (OEO) content and quality from various germplasm resources are different. Even under the same growth environment, the chemical composition of OEOs from different populations has notable differences [14,15]. In addition to the different plant types, the composition of plant essential oils is affected by various factors, such as the geographical environment and harvest period, thereby affecting their antibacterial properties [16,17]. With the demand for oregano, artificial cultivation and cultivar development of oregano have become more crucial.

*C. sakazakii*, a Gram-negative bacterium and foodborne opportunistic pathogen, can remain active at a low temperature (4 °C) and is resistant to desiccation [18,19]. It is highly harmful to newborns, premature infants, and elderly people with a weakened immune system. For example, it can lead to necrotizing enterocolitis, sepsis, meningitis, cerebrospinal effusion, peritoneal effusion, and intracerebral infarction. Moreover, it has a high mortality rate in infants, up to 80% [20,21]. Patients may experience chronic neurological disease and developmental complications [19]. Therefore, the International Commission for the Microbiological Specifications of Foods has classified *C. sakazakii* as a “serious hazard for restricted populations, life-threatening or with severe chronic sequelae” [22].

External stimuli, such as drying, toxic substances, and antibiotics, can make bacteria enter a viable but non-cultivable state [23,24], which may be related to the tolerance of *C. sakazakii*. Although *C. sakazakii* was removed by heat and drying processes, it still had a high detection rate in foods such as powdered infant formula [25,26,27]. Cells in the viable but non-cultivable state can still be pathogenic, posing certain public health risks [24,25]. In addition, the abuse of antibiotics enhances bacterial resistance, as evidenced by the isolation of environmental strains resistant to tetracycline, penicillin, trimethoprim, and other antibacterial drugs [28]. Therefore, finding alternatives to antibiotics, solving the current bacteriostatic dilemma, and effectively killing bacteria have become research topics for food safety [29].

OEO is a potent substitute to antibiotics, and the cultivation of oregano has received much attention. Our previous study evaluated the antibacterial activity of OEO extracted from different parts of the same site [30]. In this study, the antibacterial properties of OEOs from eight oregano varieties were evaluated by comparing the diameter of the inhibition zone (DIZ), the minimum inhibitory concentration (MIC), and the minimum bactericidal concentration (MBC). In addition, the effects of thymol, the main antibacterial compound of OEO, on *C. sakazakii* cell membrane integrity and morphology, protein leakage concentration, and intracellular ATP concentration were examined.

## 2. Results and Discussion

### 2.1. OEO Compounds of Eight Oregano Varieties

Sixty-seven compounds were detected in OEOs from the eight oregano varieties by GC–MS (Table 1). The total proportion of detected compounds was 95.35–99.13%. Sabinene, p-cymene, γ-terpinene, methyl thymyl ether, thymol, carvacrol, and spathulenol, mainly terpenes, were largely present in OEOs. Ovsc1 and Ovsc2 were similar in their main compounds (Table 1). Thymol was the most abundant constituent of OEOs, with a relative percentage of 53.95%, which was consistent with previous results [31]. The main OEO compounds of Ovny, Oveh, and Ovvr were similar, and the top three compounds were thymol, γ-terpinene, and p-cymene separately. All OEO compounds were visualized using a heatmap (Figure 1A). According to the plant materials and the composition analysis results, habitats may have specific effects on OEO composition, in addition to various species, artificial cultivation, and selection factors [2,32,33].

### 2.2. Clustering and Diversification Analysis of OEO Composition

The principal component analysis is shown in Figure 1B. They were clustered into four categories according to their compounds, which can be observed from the nodes in the figure. In addition, carvacrol, γ-terpinene, and p-cymene had a negative correlation, which was determined by analyzing each vector (Figure 1B). In contrast, p-cymene and thymol had a positive correlation. The correlation between these compounds may be related to the anabolic pathway of essential oils in plants. Furthermore, based on the dendrogram results, all eight OEOs could be grouped into the following two categories: group 1 (red line) of plant material was collected from the wild, and group 2 (blue line) was precisely collected from Institute of Botany, Chinese Academy of Sciences (Figure 1C). According to OEO composition, the relative content of thymol was higher in wild oregano than in cultivated oregano, which was consistent with the findings of El Gendy et al. [34]. In future oregano cultivar development, the wild-type with high relative content of thymol in OEO can be considered parents to obtain valuable cultivars.

Comprehensive data analysis of 24 samples (Figure 1) showed that terpenes were the main OEO compounds. Furthermore, group 1 contained more monoterpenes, and group 2 contained more derivatives of terpenes. OEOs in group 1 were significantly different from those in group 2, in terms of composition, which may be an observable target in oregano breeding and one piece of evidence for oregano-related functional changes.

### 2.3. Orthogonal Partial Least Squares Discriminant Analysis (OPLS-DA) of Eight OEOs

All samples were divided into two groups according to the dendrogram results (Figure 1C). A supervised OPLS-DA statistical method was used to explore the compound clustering (Figure 2). The main characteristic compounds in group 1 were methyl thymyl ether, spathulenol, and α-terpinene. Group 2 was characterized by sabinen, caryophyllene, germacrene D, and (+)-4-carene. In the variable importance projection (VIP) diagram (Figure 2C), 3-thujene, (+)-4-carene, endo-borneol, α-terpinene, and α-phellandrene had higher VIP values. These may be the critical basis for classification. However, in this study, compounds with very low relative content were taken into consideration, which may have a particular impact on the VIP value of the compounds.

### 2.4. Analysis of Common and Unique Compounds of Eight OEOs

Upset Venn and flower diagrams were drawn to visualize the composition dataset and further compare the compounds of the eight OEOs. Based on the Upset Venn diagram (Figure 3A), the number of compounds of the 8 OEOs ranged from 15 to 38. Except for Ovxj, the other compounds of oregano collected from the wild were relatively simple, with fewer than 20 compounds. However, the oregano composition in group 2 was relatively balanced, and it was more complex than that in group 1. There were six components shared by eight OEOs (Figure 3B), and only one of them was shared by oregano in group 1.

According to the stacked graph of the standard compound relationships among the eight oregano varieties, the six compounds shared by all OEOs were 3-thujene, p-cymene, γ-terpinene, thymol, carvacrol, and caryophyllene (Figure 3C). Thymol accounted for the largest proportion in Ovjx and Ovsc2; p-cymene, γ-terpinene, and thymol accounted for equal importance in Ovny, Oveh, and Ovvr. These six compounds were similar in proportion in Ovvl and Ovxj. Overall, among the six compounds shared by the eight OEOs, thymol accounted for the largest proportion and varied the most. Eleven compounds were shared by oregano in group 2 (Figure 3D), which were similar in terms of the relative content. Furthermore, the stacked graph showed that the relative proportions of (+)-4-carene and β-phellandrene were high.

These results demonstrated that the OEO chemical compounds of both groups 1 and 2 were similar and the distribution of antibacterial compounds was uniform. Nevertheless, the relative percentages of these compounds were significantly different. Compared with group 1, the OEO compounds that belonged to group 2 were complex.

The OEO compounds in different varieties were remarkably similar to the genetic relationship in the parents selected for cultivation. They may also be related to the consistency of the habitats. Several studies have indicated that environmental factors, such as altitude, water availability, and pedo-climatic conditions, affect the proportion of compounds and qualitative chemical composition [35,36]. Perhaps because of the certain altitude and climatic conditions in Xinjiang, eight unique OEO compounds were detected in Ovxj. 

### 2.5. Antibacterial Activity of OEOs

DIZ, MIC, and MBC were measured to evaluate the antibacterial properties of the eight OEOs (Figure 4). All eight OEOs had potent antibacterial activity, with 1–8 mg/mL of MIC and MBC values. DIZ of OEO was mostly above 10 mm, which was consistent with the high-efficiency inhibitory effect results of OEO on *E. coli*, *S. aureus*, *L. monocytogenes*, and other foodborne pathogens [37,38,39]. DMSO had no inhibitory effect on *C. sakazakii*, and the inhibition zone was almost absent (Figure 4C).

The DIZ results showed that the antibacterial activity of OEOs against *C. sakazakii* can be divided into the following three levels: (1) Ovsc2 was the strongest, reaching 18.22 mm; (2) Ovsc1, Ovjx, Ovny, and Oveh were moderate; and (3) Ovvl and Ovxj were the weakest, which showed no significant difference from DMSO. The antibacterial properties of OEOs from group 2 were relatively average, and most were at a medium level. In contrast, group 1 oregano had excellent or inferior antibacterial properties, which were relatively scattered, suggesting that the cultivation environment impacted the OEO composition, thereby affecting its antibacterial activity.

The MIC and MBC values of OEOs were relatively consistent with the trend of DIZ results (Figure 4A,B). Similar to the DIZ result, the OEOs of Ovvl and Ovxj had the highest MIC and MBC values, implying that their antibacterial ability was the weakest. However, the OEO of Ovny, which belonged to group 2, had the lowest MBC value. Figure 4 showed that the MIC and MBC values of group 2 OEOs were more potent than those of group 1. The different ductility of OEOs might affect their performance in the DIZ test, which might be the reason for the slight deviation from the DIZ results.

On the basis of the antibacterial status of OEOs and the proportional distribution of each compound, it was determined that thymol, one of the compounds shared by all OEOs (Figure 3C), was the most variable compound. In addition, DIZ of OEO was consistent with the variation in thymol concentration (Figure 4C). After studying the relation between the thymol content in OEOs and antibacterial activity, it was found that the greater the thymol concentration, the stronger the OEO antibacterial impact, and a substantial positive correlation was observed between the two variables (Figure 4D). The antibacterial activity of OEO is produced by the synergistic action of various chemical compounds; thus, it is not completely positively correlated with the relative content of thymol. Essential oil and its main antibacterial compounds do not easily produce drug resistance because of multiple targets. Therefore, they are ideal natural antibacterial agents to replace antibiotics [40]. However, clarifying the mechanism of action of the core antibacterial compounds is essential for the development of OEO as an alternative to antibiotics. Thus, the specific antibacterial activity and mechanism of thymol, the main compound in OEOs, require further study. 

### 2.6. Antibacterial Activity of Thymol against C. sakazakii

CLSM was used to detect cell membrane damage (Figure 5A). With the increase in thymol concentration, the green fluorescence gradually decreased and the red fluorescence gradually increased. SEM can observe changes in cell morphology, which was direct evidence that thymol may disrupt bacterial structures. Bacteria in the control group had a smooth surface and a complete morphology (Figure 5B). When the concentration of thymol reached ½ MIC, the morphology of bacteria was basically complete. In Figure 5C–E, it can be observed that the protein leakage occurred at this time. However, *C. sakazakii* treated with ½ MIC of thymol could still maintain certain normal life activities compared with those treated with MIC of thymol.

With the increase in thymol concentration, the bacterial surface appeared to fold, and when the treatment concentration reached the MIC value, the bacterial membrane was completely damaged. The concentration of extracellular proteins and nucleic acids significantly increased with thymol concentration (Figure 5C,D), indicating that the severity of bacterial membrane damage was gradual. This further substantiated that treatment with thymol disrupted bacterial membranes. In addition to changes in bacterial membranes, thymol severely impacted the normal life activities of *C. sakazakii*. Compared with the control, ATP concentration in bacteria significantly decreased after treatment, indicating that thymol treatment had a significant inhibitory effect on the normal life activities of *C. sakazakii* (Figure 5E).

Previous studies have shown that the antibacterial mechanism of carvacrol, an isomer of thymol, works mainly by disrupting bacterial membranes, affecting the leakage of bacterial intracellular components, and the normal metabolic activities of bacteria, resulting in decreased bacterial activity and even death [3,41]. In the present study, thymol was found to destroy bacterial membranes to produce antibacterial effects, indicating that the antibacterial mechanisms of thymol and carvacrol had certain similarities. Furthermore, it provided a specific theoretical basis for the similar application of different OEO compounds. Further studies will explore differences in the antibacterial mechanism between thymol and carvacrol at the molecular level. 

## 3. Materials and Methods

### 3.1. Bacterial Strain

*C. sakazakii* ATCC 29544 was preserved in 25% glycerol at −80 °C at the Institute of Botany, Chinese Academy of Sciences. Before use, cells of the strain were taken out and cultured in Luria–Bertani (LB) broth at 37 °C with 180 rpm for 12 h.

### 3.2. Plant Material and Extraction of OEOs

Details of the plant materials used in this study are provided in Table 2.

The specimens were kept at the Chinese National Herbarium, Institute of Botany, Chinese Academy of Sciences, with the following accession numbers: Ovjx (PE 02347462), Ovxj (PE 02347461), Ovsc1 (PE 02347445), Ovsc2 (PE 02347444), Ovvl (PE 02347464), and Ovvr (PE 02347463). Ovny and Oveh are new varieties cultivated by our research group, with patent numbers 20210721 and 20210717. All plant materials were taken from aerial parts and dried at room temperature in the shade. Samples were then ground to a powder. The powdered samples (100 g) were mixed with 1000 mL of distilled water, and a steam distillation method with a Clevenger apparatus was used to extract the OEOs. After boiling, the extraction process was performed for 3 h. Extracted OEOs were dried using anhydrous sodium sulfate and stored in an amber bottle at 4 °C [30,31].

### 3.3. Analysis of OEO Composition

All OEO samples were filtered and diluted with n-hexane at a 1:200 ratio. The 7890A-7000B GC–MS (Agilent Technologies, Santa Clara, CA, USA), equipped with an HP-5MS column (30 m × 250 μm × 0.25 μm; Agilent Technologies), was used to perform gas chromatography–mass spectrometry (GC–MS), and the specific procedure was a modified method described previously [42]. The injector temperature was 250 °C. The temperature program was as follows: OEO sample (1 μL) was injected in the split mode at 20:1; the temperature was maintained at 60 °C for 4 min and was then increased linearly to 84 °C at a rate of 6 °C/min; the temperature was then increased to 102 °C at a rate of 4 °C/min, next increased to 156 °C at a rate of 6 °C/min, followed by increasing to 180 °C at 4 °C/min. Finally, the temperature was increased to 280 °C at a rate of 20 °C/min. The transfer line temperature was 280 °C, and helium was used as the carrier gas at a flow rate of 1.2 mL/min through the column. The MS conditions were as follows: ionization energy, 70 eV; electronic impact ion source temperature, 230 °C; quadrupole temperature, 150 °C; and mass range, 40–700 μL. Simultaneously, the relevant information for n-alkanes was determined, and the relative percentage of each compound was determined according to the peak area. The retention index (RI) value of each compound was calculated according to the RI value and C7-C40 data, and compared with the NIST 17 library to determine the specific compound of each OEO [43].

### 3.4. Antibacterial Activity of Extracted OEOs

#### 3.4.1. Diameter of Inhibition Zone (DIZ) Measurement

The disc diffusion method, with some modifications, was used to compare the zone of inhibition of OEOs and to evaluate their antibacterial activity in one dimension. Then, 100 μL of *C. sakazakii* bacterial suspension (10^7^ CFU/mL) was spread evenly on an LB agar plate, and sterilized special filter paper with a diameter of 6 mm was placed in the center of the plate, and 10 μL of 200 mg/mL OEO diluted with dimethylsulfoxide (DMSO) was carefully dropped onto the plate. The plates were incubated at 37 °C for 24 h, and the DIZ value was measured with a Vernier caliper (AIRAJ, Qingdao, China). Ampicillin (10 μg) was the positive control, and DMSO was the negative control.

#### 3.4.2. Minimum Inhibitory Concentration (MIC) and Minimum Bactericidal Concentration (MBC) Assays

MIC and MBC of *C. sakazakii* were assessed according to our previous method [42]. OEOs were diluted with LB broth containing 10^7^ CFU/mL at two-fold serial dilutions to obtain concentrations of 16, 8, 4, 2, 1, 0.5, 0.25, and 0.125 mg/mL. Bacterial suspensions without OEO were used as the control group. The samples were then incubated for 24 h at 37 °C. The lowest concentration of OEO at which the medium had no visible bacterial growth was considered the MIC value. After MIC determination, bacterial suspensions that contained an OEO (not less than MIC) were spread evenly on agar. The lowest concentration of OEO at which the medium excluded visible bacterial colonies was considered the MBC value. The MIC and MBC values of thymol were determined in the same way as OEOs.

### 3.5. Cell Membrane Integrity Assay

*C. sakazakii* suspensions (~10^7^ CFU/mL) were treated for 4 h with or without thymol at final concentrations of 0, ½ MIC, and MIC. Then, the treated *C. sakazakii* cells were washed three times with phosphate-buffered saline (PBS) and stained with 5 µM SYTO9 and 15 µM propidium iodide (PI) (Molecular Probes, Invitrogen, France) in the dark for 15 min [44,45]. SYTO9 and PI were diluted with 0.85% saline. Next, cells were washed three times with PBS and resuspended in PBS before analysis. The cell membrane integrity was measured using a confocal laser scanning microscope (CLSM; Zeiss LSM 980 with Elyra7, Jena, Germany).

### 3.6. Scanning Electron Microscope (SEM) Observations of C. sakazakii

Changes in morphological characteristics of *C. sakazakii* after exposure to thymol were observed as described by Hao et al. [45], with some modifications. *C. sakazakii* ATCC 29544 (10^7^ CFU/mL) was treated with thymol at final concentrations of 0, ½ MIC, and MIC at 37 °C for 4 h. The samples were then centrifuged (5000× *g*, 10 min, 4 °C) and washed three times with PBS. Sterile water containing 2.5% glutaraldehyde was used to fix cells for 6 h at 4 °C. Then, the *C. sakazakii* cells were dehydrated using a water–ethanol gradient (25%, 50%, 75%, 90%, 95%, 100%), with 10 min standing for each concentration. After dehydration and drying, the samples were fixed on the SEM support and sprayed with gold by ion sputtering. Finally, all samples were examined using a SEM (Regulus 8100, Hitachi Co., Ltd., Japan) [46]. 

### 3.7. Bacterial Extracellular Protein Assay

The extracellular protein concentration of the samples was detected with a micro BCA protein assay kit (C503061, Sangon Biotech, Shanghai, China). The standard curve assay solution was prepared according to the kit instructions. The bacterial solution treated with thymol at final concentrations of 0, ½ MIC, and MIC was centrifuged at 5000× *g* for 5 min, and the supernatant was diluted to a specific multiple. A microplate reader (Thermo Fisher Scientific, Waltham, MA, USA) was used to measure the standard and test solutions at 562 nm. Extracellular protein concentration of the samples was calculated according to the standard curve.

### 3.8. Bacterial Extracellular Nucleic Acid Assay

The treatment approach of *C. sakazakii* suspensions was identical to the pretreatment procedure of extracellular protein concentration measurement. After collecting the supernatant, the extracellular nucleic acid concentration was determined using a nucleic acid concentration detector (IMPLEN, Munich, Germany).

### 3.9. Bacterial Intracellular ATP Assay

To determine the ATP content of cells with different treatments, the residue, collected after centrifugation at 5000× *g* for 5 min, was detected using an ATP bioluminescence assay kit (S0026, Beyotime, Haimen, China), following the manufacturer’s instructions. A standard ATP concentration curve was used to determine the intracellular ATP concentration of *C. sakazakii* suspensions.

### 3.10. Statistical Analysis

All experiments were independently performed in triplicate. The difference in DIZ (*p* < 0.05 was considered significant) was determined using SPSS software (version 25.0; IBM, Endicott, NY, USA). The OEO compounds and their correlation with antibacterial activity were analyzed by heatmaps, principal component analysis, cluster analysis, UpSet Venn, and flower diagrams based on the R drawing platform. Correlations between DIZ and OEO compounds were analyzed using Origin 2021 (OriginLab, Northampton, MA, USA).

## 4. Conclusions

OEOs had potent antimicrobial activity against *C. sakazakii*. Through multidimensional exploration, oregano collected from different environments showed significant differences in OEO composition. Moreover, the relative content of thymol was positively correlated with OEO antibacterial activity, which implied that thymol was important to the biological activities of OEOs. However, the correlation was not always positive between them, which may be because of the synergistic and antagonistic effects among the various compounds. In addition, as the main antibacterial compound in these OEOs, thymol appeared to have some definitive mechanisms of action on *C. sakazakii*, including decreasing bacterial activity, disrupting the bacterial membrane, causing the leakage of bacterial proteins, and reducing the intracellular ATP concentration. This study evaluated the antibacterial properties of OEOs and provided evidence for the antibacterial activity of thymol. Moreover, it provided a theoretical foundation for oregano cultivar development from wild oregano resources for OEO with a high relative content of thymol.

## Figures and Tables

**Figure 1 molecules-27-06702-f001:**
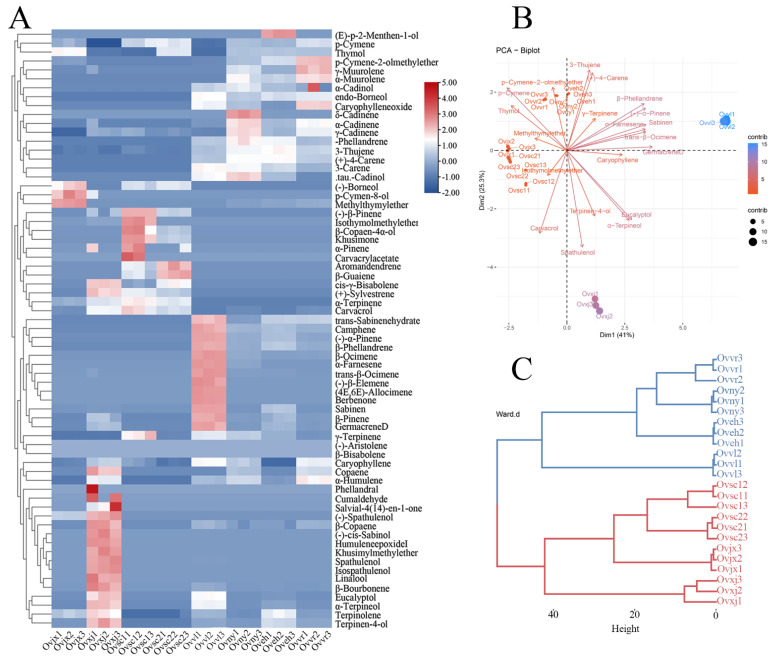
Composition analysis of OEOs. The results are illustrated by a heatmap (**A**), PCA plot (**B**), and dendrogram (**C**).

**Figure 2 molecules-27-06702-f002:**
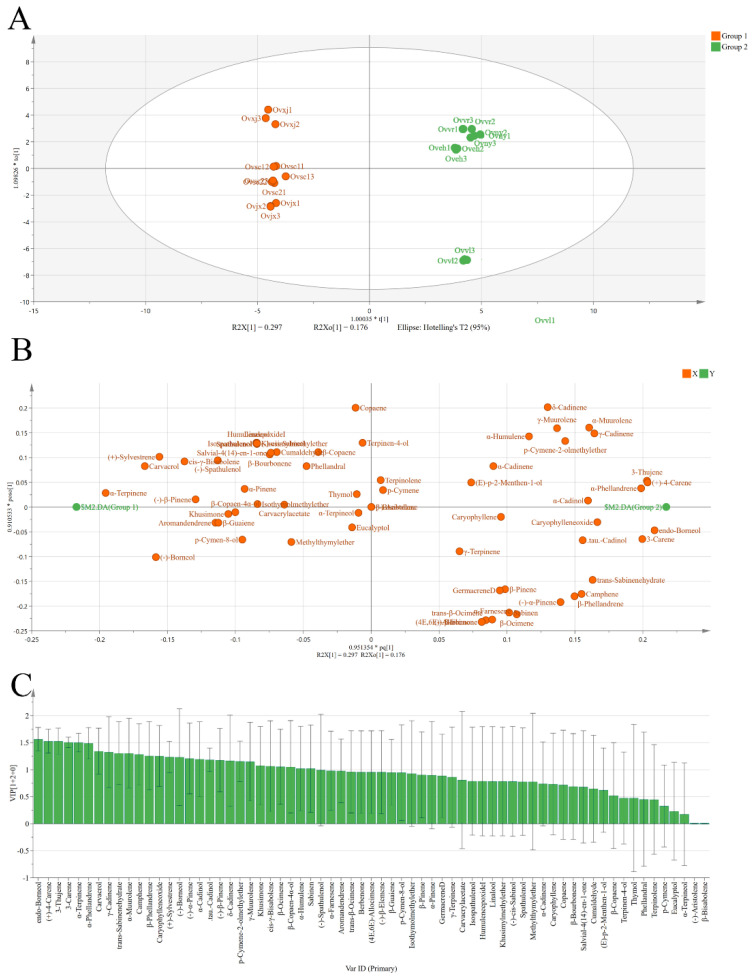
Score plots (**A**), loading plot (**B**), and VIP values (**C**) from OPLS-DA based on the chemical profiles of eight OEOs.

**Figure 3 molecules-27-06702-f003:**
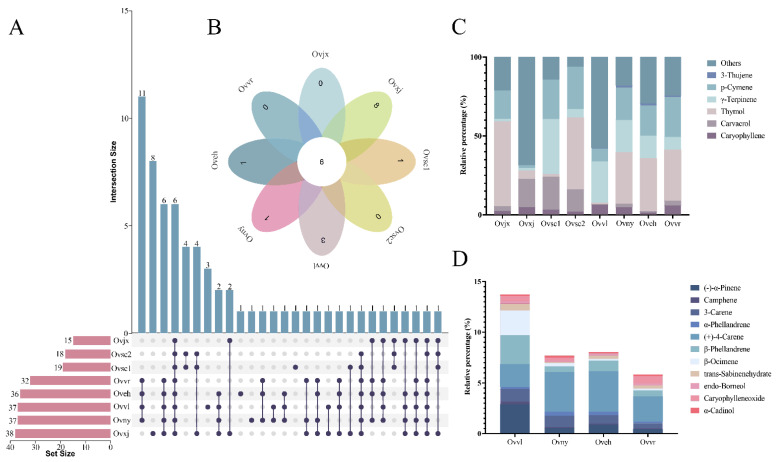
Distribution of common and unique compounds. Amounts of OEO chemical compounds illustrated by an UpSet Venn diagram (**A**) and flower diagram (**B**). Stacked graph (**C**) of common compounds in all samples. Stacked graph (**D**) of shared compounds in all cultivated varieties.

**Figure 4 molecules-27-06702-f004:**
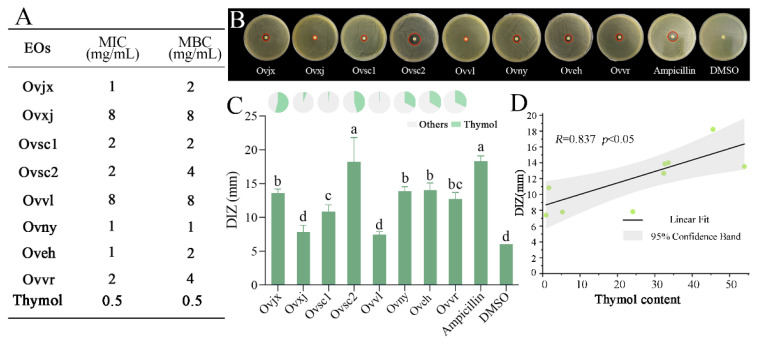
Antibacterial activity of OEOs. Minimum inhibitory concentration (MIC) and minimum bactericidal concentration (MBC) (**A**), the diameter of inhibition zone (DIZ) images (**B**). Statistical data (histogram in **C**) for OEOs against *C. sakazakii*, and percentages of thymol in the eight OEOs (pie graph in **C**); Discs were 6 mm in diameter, and values are means ± standard deviations; different lowercase letters (a, b, c, d) above bars indicate significant differences (*p* < 0.05) (**C**), correlation between thymol content in OEOs and its antibacterial activity against *C. sakazakii* (**D**).

**Figure 5 molecules-27-06702-f005:**
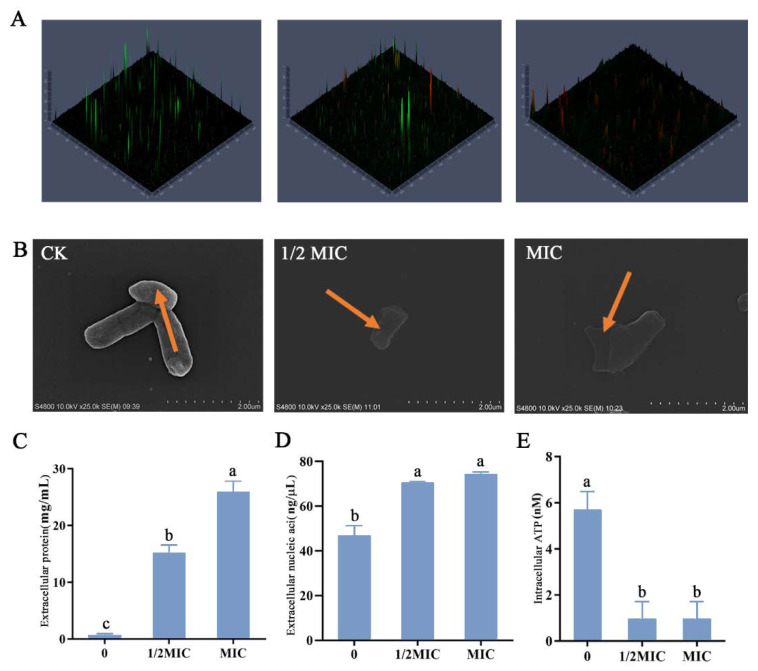
Cell damage of *C. sakazakii* ATCC 29544. CLSM showed *C. sakazakii* viability treated with different concentrations of thymol. (**A**) Cells stained with propidium iodide (PI) are labeled red, whereas cells stained with SYTO9 are labeled green. (**B**) Scanning electron micrographs of *C. sakazakii* ATCC 29544 treated with different concentrations of thymol. (**C**) Extracellular protein concentration. (**D**) Extracellular nucleic acid concentration. (**E**) Intracellular ATP concentration. Different lowercase letters (a, b, c) above bars indicate significant differences (*p* < 0.05).

**Table 1 molecules-27-06702-t001:** Chemical composition of essential oils from eight oregano varieties.

Compound	RI	Relative Concentration (%)							
		Ovjx	Ovxj	Ovsc1	Ovsc2	Ovvl	Ovny	Oveh	Ovvr
3-Thujene	928	0.07 ± 0.06	0.04 ± 0.07	0.23 ± 0.06	0.11 ± 0.01	0.83 ± 0.02	1.35 ± 0.01	1.48 ± 0.04	1.02 ± 0.03
α-Pinene	932	-	0.04 ± 0.07	0.11 ± 0.09	0.03 ± 0.05	-	-	-	-
(-)-α-Pinene	934	-	-	-	-	2.9 ± 0.04	0.57 ± 0.02	0.89 ± 0.02	0.45 ± 0.02
Camphene	949	-	-	-	-	0.27 ± 0.01	0.07 ± 0	0.09 ± 0	0.07 ± 0
Sabinen	972	**0.08 ± 0.07 ^d^**	**0.53 ± 0.06 ^c^**	-	-	**14.97 ± 0.08 ^a^**	**0.06 ± 0 ^d^**	**4.6 ± 0.2 ^b^**	-
β-Pinene	976	-	0.32 ± 0.11	-	-	1.02 ± 0.03	0.12 ± 0.01	0.27 ± 0.06	0.11 ± 0.01
(-)-β-Pinene	990	-	0.13 ± 0.11	0.54 ± 0	0.23 ± 0.01	-	-	-	-
3-Carene	992	-	-	-	-	1.24 ± 0.03	1.14 ± 0.02	0.86 ± 0.05	0.5 ± 0.01
α-Phellandrene	1004	-	-	-	-	0.22 ± 0	0.4 ± 0.03	0.34 ± 0.01	0.18 ± 0.01
α-Terpinene	1016	0.48 ± 0.04	1.01 ± 0.04	1.36 ± 0.04	1.06 ± 0.05	-	-	-	-
(+)-4-Carene	1017	-	-	-	-	2.23 ± 0.09	3.88 ± 0.03	3.96 ± 0.08	2.47 ± 0.05
p-Cymene	1024	**17.93 ± 1.53 ^d^**	**1.56 ± 0.04 ^f^**	**24.78 ± 2.38 ^b^**	**26.8 ± 1.1 ^a^**	**7.76 ± 0.25 ^e^**	**20.45 ± 0.25 ^c^**	**19.05 ± 0.34 ^c^**	**25.22 ± 0.45 ^a^**
β-Phellandrene	1029	-	-	-	-	2.84 ± 0.04	0.6 ± 0.02	1.09 ± 0.03	0.59 ± 0.02
(+)-Sylvestrene	1029	-	0.38 ± 0.03	0.26 ± 0.01	0.17 ± 0.01	-	-	-	-
Eucalyptol	1031	-	**7.23 ± 0.61 ^a^**	-	-	**5.61 ± 0.18 ^b^**	**0.09 ± 0 ^c^**	**0.41 ± 0.01 ^c^**	-
β-Ocimene	1037	-	-	-	-	2.44 ± 0.06	0.33 ± 0.01	0.15 ± 0	0.18 ± 0.01
*trans*-β-Ocimene	1047	-	0.06 ± 0.11	-	-	4.27 ± 0.12	0.09 ± 0	0.09 ± 0	-
γ-Terpinene	1059	**1.78 ± 0.02 ^f^**	**1.99 ± 0.08 ^f^**	**35.1 ± 4.54 ^a^**	**5.49 ± 0.17 ^e^**	**26.34 ± 0.35 ^b^**	**20.61 ± 0.39 ^c^**	**14.43 ± 0.06 ^d^**	**8.19 ± 0.13 ^e^**
*trans*-Sabinenehydrate	1066	-	-	-	-	0.68 ± 0.03	0.1 ± 0	0.31 ± 0.01	0.32 ± 0.01
Terpinolene	1089	0.33 ± 0.03	0.55 ± 0.05	-	0.03 ± 0.05	0.26 ± 0	0.12 ± 0	0.45 ± 0.03	0.18 ± 0
Linalool	1100	-	1.84 ± 0.24	-	-	-	-	-	-
(4*E*,6*E*)-Allocimene	1130	-	-	-	-	0.13 ± 0	-	-	-
(-)-*cis*-Sabinol	1141	-	0.27 ± 0.03	-	-	-	-	-	-
(*E*)-p-2-Menthen-1-ol	1141	-	-	-	-	-	-	0.07 ± 0	-
(-)-Borneol	1168	0.25 ± 0.03	-	0.15 ± 0.02	0.17 ± 0.01	-	-	-	-
*endo*-Borneol	1169	-	-	-	-	0.24 ± 0	0.14 ± 0	0.18 ± 0.01	0.2 ± 0
Terpinen-4-ol	1179	0.47 ± 0.01	3.54 ± 0.36	-	-	0.83 ± 0.02	0.53 ± 0	1.84 ± 0.09	0.8 ± 0.01
p-Cymen-8-ol	1186	0.23 ± 0.04	0.04 ± 0.07	-	-	-	-	-	-
α-Terpineol	1192	-	2.85 ± 0.19	-	-	1.95 ± 0.03	0.1 ± 0	0.46 ± 0	0.13 ± 0
Berbenone	1211	-	-	-	-	0.18 ± 0	-	-	-
Methylthymylether	1237	**16.3 ± 0.5 ^a^**	-	-	-	-	**1.17 ± 0.05 ^c^**	**1.63 ± 0.07 ^c^**	**2.41 ± 0.1 ^b^**
Cumaldehyde	1242	-	0.13 ± 0.12	-	-	-	-	-	-
Isothymolmethylether	1245	-	**0.47 ± 0.03 ^b^**	**7.11 ± 0.67 ^a^**	-	-	-	-	-
p-Cymene-2-olmethylether	1246	**0.98 ± 0.07 ^d^**	-	-	-	**0.2 ± 0.01 ^e^**	**2.39 ± 0.07 ^c^**	**3.57 ± 0.07 ^b^**	**6.48 ± 0.15 ^a^**
Phellandral	1278	-	0.03 ± 0.05	-	-	-	-	-	-
Thymol	1292	**53.95 ± 1.68 ^a^**	**5.26 ± 0.07 ^e^**	**1.64 ± 0.44 ^f^**	**45.56 ± 0.18 ^b^**	**0.92 ± 0.03 ^f^**	**32.71 ± 0.43 ^cd^**	**33.62 ± 0.65 ^c^**	**32.39 ± 0.11 ^d^**
Carvacrol	1305	**2.92 ± 0.31 ^d^**	**17.88 ± 0.28 ^b^**	**20.96 ± 3.96 ^a^**	**14.15 ± 0.73 ^c^**	**0.43 ± 0.02 ^e^**	**2.23 ± 0.02 ^de^**	**0.83 ± 0.06 ^de^**	**3.04 ± 0.19 ^d^**
Carvacrylacetate	1374	-	-	0.09 ± 0.08	-	-	-	-	-
Copaene	1381	-	0.22 ± 0.03	-	-	-	0.08 ± 0	-	0.11 ± 0
β-Bourbonene	1390	-	3.9 ± 0.44	-	-	0.43 ± 0.07	-	0.04 ± 0.04	0.02 ± 0.04
(-)-β-Elemene	1395	-	-	-	-	0.17 ± 0.01	-	-	-
Caryophyllene	1425	**2.4 ± 0.11 ^e^**	**4.85 ± 0.25 ^c^**	**3.18 ± 0.41 ^d^**	**1.96 ± 0.14 ^f^**	**6.36 ± 0.11 ^a^**	**4.68 ± 0.24 ^c^**	**1.39 ± 0.02 ^g^**	**5.88 ± 0.07 ^b^**
β-Copaene	1434	-	0.59 ± 0.05	-	-	0.13 ± 0.02	0.04 ± 0.04	0.04 ± 0.03	0.13 ± 0.01
Aromandendrene	1445	-	-	0.36 ± 0.1	0.69 ± 0.08	-	-	-	-
α-Humulene	1459	-	0.73 ± 0.02	-	-	0.43 ± 0.02	0.61 ± 0.01	0.23 ± 0.01	0.87 ± 0.06
(-)-Aristolene	1466	-	-	-	-	-	-	-	-
γ-Muurolene	1481	-	0.04 ± 0.06	-	-	-	0.3 ± 0.01	0.13 ± 0	0.56 ± 0.04
GermacreneD	1486	-	**2.07 ± 0.13 ^b^**	-	-	**6.57 ± 0.32 ^a^**	**0.59 ± 0.02 ^d^**	**1.73 ± 0.07 ^c^**	**0.66 ± 0.07 ^d^**
β-Guaiene	1500	-	-	0.38 ± 0.13	0.83 ± 0.06	-	-	-	-
α-Muurolene	1505	-	-	-	-	-	0.13 ± 0.03	0.09 ± 0.01	0.17 ± 0.01
α-Farnesene	1510	-	-	-	-	4.32 ± 0.06	0.33 ± 0.02	-	-
*cis*-γ-Bisabolene	1511	-	1.43 ± 0.03	0.19 ± 0.08	1.07 ± 0.12	-	-	-	-
β-Bisabolene	1511	-	-	-	-	-	-	-	-
γ-Cadinene	1519	-	0.06 ± 0.11	-	-	0.07 ± 0	0.56 ± 0.03	0.22 ± 0.01	0.46 ± 0.02
δ-Cadinene	1527	-	0.49 ± 0.04	0.45 ± 0.12	0.34 ± 0.04	0.24 ± 0.02	1.18 ± 0.02	0.57 ± 0.03	0.94 ± 0
α-Cadinene	1542	-	-	-	-	-	0.06 ± 0	-	-
β-Copaen-4α-ol	1583	-	-	1.64 ± 0.49	0.51 ± 0.06	-	-	-	-
(-)-Spathulenol	1583	0.95 ± 0.03	2.62 ± 0.2	-	-	-	-	-	-
Khusimone	1589	-	-	0.86 ± 0.11	0.31 ± 0.03	-	-	-	-
Spathulenol	1589	-	**27.16 ± 1.86 ^a^**	-	-	**0.26 ± 0.04 ^b^**	-	-	-
Caryophylleneoxide	1591	-	-	-	-	0.53 ± 0.02	0.29 ± 0.01	0.07 ± 0	0.71 ± 0.01
Salvial-4(14)-en-1-one	1600	-	0.31 ± 0.21	-	-	-	-	-	-
HumuleneepoxideI	1615	-	3.5 ± 0.19	-	-	-	-	-	-
Isospathulenol	1641	-	0.23 ± 0.01	-	-	-	-	-	-
.tau.-Cadinol	1645	-	-	-	-	0.18 ± 0.01	0.24 ± 0.01	0.09 ± 0	-
α-Cadinol	1660	-	-	-	-	0.14 ± 0.01	0.17 ± 0.01	0.1 ± 0	0.16 ± 0.23
Khusimylmethylether	1675	-	1.3 ± 0.12	-	-	-	-	-	-
Total		99.13 ± 0.24	95.65 ± 1.16	99.38 ± 0.16	99.52 ± 0.03	98.58 ± 0.05	98.52 ± 0.13	95.35 ± 0.12	95.59 ± 0.34

**Notes**: RI, retention indices; values are presented as mean ± standard deviation of three parallel experiments; ‘-’ means not detected. Means with different letters in a row are statistically significant (*p* < 0.05).

**Table 2 molecules-27-06702-t002:** Plant materials and collection sites.

Plant	Collection Date	Abbreviation	Collection Site
*Origanum vulgare* jx	19 August 2019	Ovjx	YuShan County, Jiangxi, China
*O. vulgare* xj	1 July 2020	Ovxj	Xinjiang, China (81°36′39′′ N, 44°6′23.58′′ E)
*O. vulgare* sc1	6 July 2019	Ovsc1	The top of Gongga Mountain in Sichuan, China
*O. vulgare* sc2	10 July 2019	Ovsc2	The foot of Gongga Mountain in Sichuan, China
*O. vulgare* ssp. *vulgare*	21 June 2020	Ovvl	Institute of Botany, Chinese Academy of Sciences, Beijing, China (39°48′ N, 116°28′ E)
*O.* ‘Nvying’	25 June 2020	Ovny	Institute of Botany, Chinese Academy of Sciences, Beijing, China (39°48′ N, 116°28′ E)
*O. vulgare* ‘Ehuang’	25 June 2020	Oveh	Institute of Botany, Chinese Academy of Sciences, Beijing, China (39°48′ N, 116°28′ E)
*O. vulgare* ssp. *virens*	10 June 2020	Ovvr	Institute of Botany, Chinese Academy of Sciences, Beijing, China (39°48′ N, 116°28′ E)

## Data Availability

Not applicable.

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
