# Peer review of "Comparison of Origanum Essential Oil Chemical Compounds and Their Antibacterial Activity against Cronobacter sakazakii"

_molecules, 2022, doi:10.3390/molecules27196702_

Round 1
Reviewer 1 Report
The authors evaluate the antibacterial activity against Cronobacter sakazakii of 8 essential oils extracted from Oregano (both from cultivated populations and wild harvested) and evaluated the effect of a major component, thymol, for further mechanism of action(morphological changes and changes in extracellular nucleic acid, protein content and ATP) .
Major concerns:
1) An English editor is required to thoroughly edit the manuscript as there are too many sentence construction errors/ grammatical errors to list
2) Table 1 requires herbarium specimen numbers for the plant collection
3) Section 2.4.1 - controls (vehicle, untreated, positive control) should be included as well as concentrations - this should be included for all methods
4) Method of section 2.4.2 is not clear and should be re-written so as it can be repeated by other researchers
5) Section 2.5, 2.6 and 2.7 - it is not clear how the authors determined the MIC values of thymol as it was not described in the methods that it was tested for its antibacterial activity (zone of inhibition, MIC, and MBC)- this should be clearly stated
6) There is a lack in overall discussion of findings and comparison to already published literature relating to the research topic. Example elaborate more on the difference in thymol content between cultivated and wild species and how this could effect product development, etc.
7) The conclusion is more a repeat of the findings rather than making a conclusion regarding the results obtained
Minor concerns
1) Section 2.4.1 - this should be changes to "zone of inhibition" not "diameter of inhibitory zone" and this should be changed throughout where applicable
2) Section 2.4.1 - why were the essential oils tested at such a high concentration of 200mg/mL - one would expect at this concentration that there will be inhibition of bacterial growth
3) Section 2.5 - what were the stains diluted in?
4) Section 2.5 and 2.6 - how was staining done? was it performed in multi-well plates?
5) Section 2.7 and 2.8 needs to specify that it was thymol being tested
6) Section 2.10 - how many replicates were performed in each independent experiment?
7) In line 291 the authors discuss a potential synergistic activity of thymol with other components. Should the three main components not have been tested in combination to observe the effect?
Author Response
Dear reviewer,
Re: Manuscript ID: molecules-1919802 and Title: Comparison of oregano essential oil chemical compounds and their antibacterial activity against Cronobacter sakazakii.
Thank you very much for your kind letter, along with your constructive comments regarding our manuscript entitled “Comparison of oregano essential oil chemical compounds and their antibacterial activity against Cronobacter sakazakii” (Manuscript ID: molecules-1919802). We have substantially revised the manuscript after thoroughly considered all the comments and suggestions. Appended to this letter is our point-by-point response to the comments. Based on the instructions provided in editor’s letter, revisions in the text are shown using the “Track Changes” function. The responses to the reviewer's comments are marked in red. Please see the attachment.

Reviewer 2 Report
Dear authors,
The manuscript is quite interesting. The article writing, for non-native English-speaking readers, is rather clear. Only on line 221 "extreme relative proportions" seemed confusing to me. On the other hand, I had to expand all the figures in the text a lot. They are very small, light in color, and therefore difficult to see.
For Figure 4, I would like to ask: there is only one significant number for MBC and MIC? Also, I believe all the statistics (nice one!) could be improved with a better explanation, mainly in item 3.6, about the antibacterial action of thymol.
Author Response

(The authors gave the same response as above.)

Reviewer 3 Report
Dear authors,
This is an important submission towards Molecules. The results are novel and interesting and the manuscript is well strctured. However, some revisions have to be incorporated to improve the initial submission. Please find below my suggestions and recommendations:
1-The title is very vague and therefore it has to be revised:eight oragano essential oil? do you mean eight sub-species ? or from the same species coming from various environments?. Please be concise, informative and short when rephrasing your title,
2- Please add the scientific name for the title and the abstract. Please use abbreviation (for plant scientific name) after defining it at first mention,
3-Line 13, I see that this abbreviaiton can be better expressed as first letter of genus + first letter of species + EO (essential oil), e.g. OVEO. Please use this logic throughout the manuscript,
4- Language must be carefully revised to correct various mistakes including typo,
5- The state-of-art-of research has to be fortified by adding newly published literature on the Origanum genus (e.g: https://doi.org/10.33263/BRIAC112.93589371, etc),
6-Please explain the choice of pathogen used,
7-The authors have to add date, and phenology at the harvest of plant material. Likewise, authentification (vouchers, botanist, Lab, etc). A brief description of local climatic and edaphic data should be added as well,
8- Scientific names have to be Italic, e.g. line 123,
9-Line 174, please provide manufactuer, company, country of software used,
10-Table 2, title has to be self-explanatory. Likewise, results must be analyzed for statistical differences,
11-Fig 1B-C, and Fig 2B are unclear.
Recommendation: Accept after major revisions.
Kind regards.
Author Response

(The authors gave the same response as above.)

Round 2
Reviewer 1 Report
Thank you for attending to the comments.
One more comment is to include all controls used in the experiments at the relevant methods section (positive control, untreated control and vehicle control).
Author Response
Dear reviewer,
Re: Manuscript ID: molecules-1919802 and Title: Comparison of oregano essential oil chemical compounds and their antibacterial activity against Cronobacter sakazakii.
Thanks for your letter, along with your constructive comments regarding our manuscript. We have carefully revised the manuscript. Appended to this letter is our point-by-point response to the comments. Based on the instructions provided in editor’s letter, revisions in the text are shown using the “Track Changes” function. The responses to the reviewer's comments are marked in red. Please see the attachment.

Reviewer 3 Report
Dear authors,
The manuscript was greatly improved and therefore I recommend its publication in Molecules.
Regards.
Author Response

(The authors gave the same response as above.)
